# Alcohol drinking delays the rate of sputum smear conversion among DR-TB patients in northwest Ethiopia; A retrospective follow-up study

**Mehari Woldemariam Merid**◉*, **Atalay Goshu Muluneh, Getahun Molla Kassa**◉

Department of Epidemiology and Biostatistics, Institute of Public Health, College of Medicine and Health Sciences, University of Gondar, Gondar, Ethiopia

* mehariho19@gmail.com

## Abstract

### Background

Sputum smear microscopy is simple and feasible technique to assess the presence of acid-fast bacilli (AFB) in the respiratory tract of patients with Drug Resistance Tuberculosis (DR-TB). Conversion of sputum smear from positive to negative is considered as an interim indicator of efficacy of anti-tubercular treatment and the program effectiveness. Although evidences regarding the factors affecting the sputum smear conversion are available on drug susceptible TB patients, there is dearth of literature about smear conversion and its predictors among DR-TB patients in the study setting. Hence, shortening the time to sputum smear conversion is desirable to reduce the likelihood of mycobacterial transmission. This study has therefore aimed at estimating the median time of sputum smear conversion and to determine its predictors.

### Methods

This was a retrospective follow-up study conducted among DR-TB patients registered for second-line anti-TB treatment in the four hospitals of Amhara regional state, Northwest Ethiopia. Of all patients enrolled to DR-TB treatment in the study setting from 2010 to 2017, 436 patients have been include for this study who fulfilled the eligibility criteria. The cox proportional hazard model was fitted and the adjusted hazard ratio (AHR) with 95% confidence interval (CI) and p <0.05 was used to declare statistical significance of the variables associated with the smear conversion.

### Results

From the 436 patients with sputum smear positive at baseline, 351 (80.5%) converted sputum smear at a median time of 48 (IQR: 30–78) days. The median time of smear conversion was 59 (95% CI: 42, 74) and 44 (95% CI: 37, 54) days among patients who had and had no history of alcohol drinking, respectively. Similarly, the median time to smear conversion was 61 (95% CI: 36, 73) days among patients with comorbid conditions and 44 (95% CI: 38, 54)

**Data Availability Statement:** All relevant data supporting this study are attached as Supporting Information files.

**Funding:** The author(s) received no specific funding for this work.

**Competing interests:** The authors have declared that no competing interests exist.

**Abbreviations:** DST, Drug Susceptibility Testing; MDR-TB, Multidrug-Resistant Tuberculosis; DR-TB, Drug-Resistant Tuberculosis; RR-TB, Rifampicin-Resistance Tuberculosis; TB, Tuberculosis; SLD, Second Line ant-TB Drugs; WHO, World Health Organization; MTB, Mycobacterium Tuberculosis; RIF, Rifampicin; LPA, Line Probe Assay; IQR, Interquartile range.

days among patients with no comorbid conditions. In the multi-variable analysis, only history of alcohol consumption [**AHR = 0.66 (0.50, 0.87)**] was found to delay significantly the rate of sputum smear conversion.

## Conclusion

In our study, the median time of sputum smear conversion was with in the expected time frame of conversion. History of alcohol consumption was found to delay significantly the rate of sputum smear conversion. The DR-TB patients are strongly advised to avoid alcohol consumption.

## Background

Tuberculosis (TB) is one of the most challenging infectious diseases in the globe. It affects people of all ages. According to the 2021 global TB report, there were an estimated 1.3 million deaths among HIV-negative people, up from 1.2 million in 2019, and an additional 214 000 deaths among HIV-positive people, a small increase from 209 000 in 2019. In 2020, the number of people dying from TB increased, previous declines in the annual number of people falling ill with TB slowed, far fewer people were diagnosed and treated for TB or provided with TB preventive treatment compared with 2019 and spending on essential TB services fell [1].

Although the Sustainable Development Goal targets at ending the TB epidemic by 2030, drug-resistant TB remains to be a serious public health problem throughout the world.

According to the World Health Organization (WHO), all TB patients should be monitored during the course of anti-tuberculosis treatment to assess their response to therapy [2]. The treatment response monitoring method mainly concerns on body weight and sputum smear examination which should be done, among others, at the end of the intensive phase of treatment [2].

Conversion of sputum smear from positive to negative is considered as an interim indicator of efficacy of anti-tubercular treatment and the program effectiveness [3–5]. The treatment outcomes are only available 18–24 months after treatment starts among drug resistance tuberculosis (DR-TB) patients [5]. Hence, sputum smear negativity is a measure of good response to anti-tuberculosis treatment.

Sputum smear microscopy is simple and feasible technique to assess the presence of acid-fast bacilli (AFB) in the respiratory tract of patients with pulmonary Drug Resistance Tuberculosis (DR-TB) [6]. Sputum smear microscopy tests were performed once a month for the first 6 months and once every 2 months from that point until the end of therapy [6].

Although evidence on the time to sputum smear conversion is dearth, some literatures conducted elsewhere have reported that median time of conversion ranges from 51 to 150 days [7–11]. Patients with non-conversion or prolonged time to sputum smear conversion may require close monitoring, longer hospitalization and protracted intensive treatment which results in unfavourable treatment outcomes [12–15]. From the public health point of view, reducing the time to sputum smear conversion is an important infection control measure [16]. Hence, shortening the time to sputum conversion is desirable to reduce the likelihood of mycobacterial transmission. Moreover, sputum smear conversion is often used by clinicians to determine the duration of treatment for drug resistant tuberculosis patients.

Several factors, such as sex, age, baseline sputum grade, smoking habits, alcoholism, HIV co-infection, have been identified in previous studies as risk factors for delayed smear conversion [8, 12, 13, 16, 17].

## Method

### Study design, setting, and population

An institutional based retrospective follow up study was conducted from all DR-TB patients registered for second-line anti-TB treatment in to the four second-line anti-TB drug treatment initiating center hospitals in Amhara regional state from September 2010 and December 2017. The Treatment Initiating Centers (TICs) include the University of Gondar comprehensive specialized hospital, Debremarkos referral hospital, Borumeda referral hospital, and Woldya general hospital. These four hospital accounts more than 90% of the RR/MDR-TB in the region. The region has the second highest number of TB cases notified yearly following Oromia region in the country. Amhara regional state is the second largest and populated state from the ten regional and two town administrative states in Ethiopia. All of the patients were bacteriologically confirmed RR/MDR-TB. The diagnosis of tuberculosis and its drug resistance was made either of the GeneXpert MTB/Rif assay or Line Probe Assay (LPA) and further triangulated by culture based phenotypic drug susceptibility testing.

All DR-TB patients registered for second-line ant-TB treatment in the region were the source population whereas those patients enrolled for treatment from September 2010 to December 2017 in the selected hospitals were the study population. Patients without recorded information on their smear conversion status were excluded. Data were extracted among 436 DR-TB patient medical records (**Fig 1**).

### Data collection and quality control

Data were abstracted from the medical records using a structured data collection form. Baseline socio-demographic, behavioural, and clinical information (including sex, age, residency, marital status, education status, occupation, smoking history, history of alcohol drinking, functional status, sputum smear result, history of treatment for 1st line anti- tuberculosis, TB registration group, underlying comorbid diseases, and HIV co-infection) were obtained from

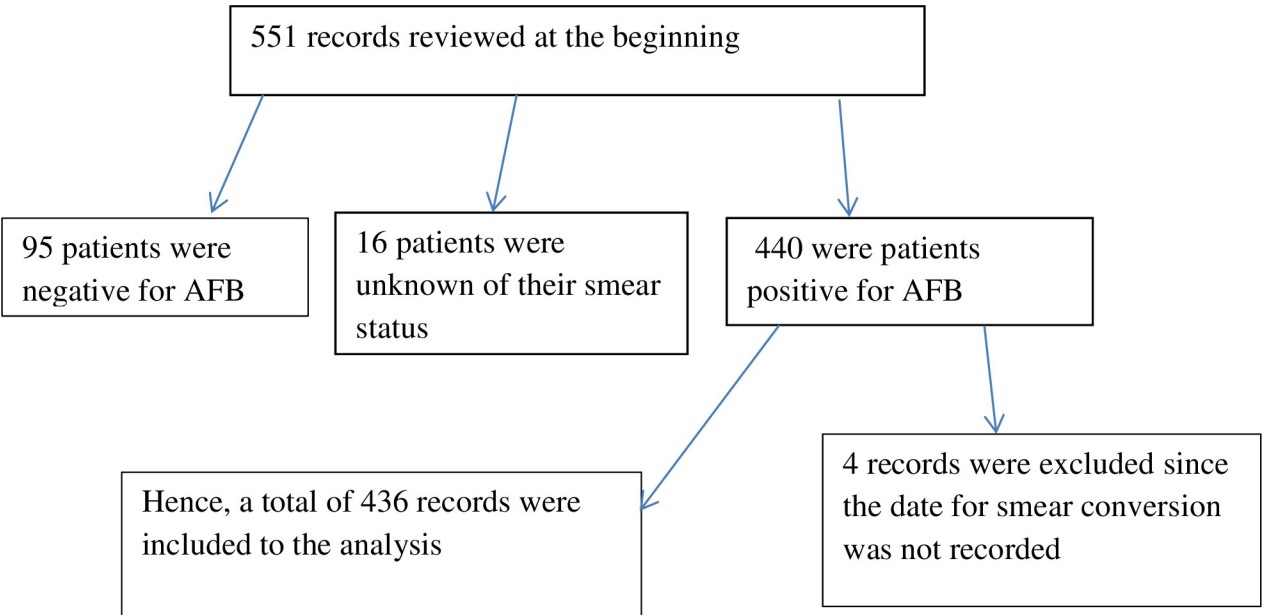

**Fig 1. The diagrammatic presentation showing a selection of DR-TB patients in Amhara Regional State Public Hospitals from September 2010 to December 2017.**

medical record reviews. In addition, we have collected and assessed the development of adverse drug events and treatment outcome. To maintain the quality of the data a half day hands-on training was given to the data collectors. Data were collected by four BSc degree students under the close supervision of the researchers. The completeness and consistence of data were inspected and corrected day-to-day.

## Study variables

Time to sputum smear conversion was the dependent variable; whereas socio-demographic characteristics (sex, age, residence, educational status, marital status), behavioural factors (smoking, alcohol drinking) and clinical characteristics (HIV co-infection, DR-TB treatment regimen, type of resistance, presence of chronic diseases, clinical complications, base line Body Mass Index (BMI), anaemia, smear grading and functional status) were the predictor variables for this study.

## Operational definitions

**Smear conversion** was defined as two consecutive negative sputum smears taken at least 30 days apart following an initial positive sputum smear [5].

   **Time to sputum Smear conversion** was defined as the time in months from the date of start of DR-TB treatment to the date of smear conversion for the first of two consecutive negative sputum smear results [5].

   **Censored** was defined as when smear result has not been converted for an individual. This includes when the patient stops the treatment though the smear was positive, deceased before culture conversion, transferred out to another treatment site before culture conversion, and study time completion before culture conversion.

   **Anemia:** was defined as hemoglobin value of less than 12 mg/dl for males and less than 11 mg/dl for females [18]. Sputum smear was graded as scanty, 1+, 2+, and 3+. Scanty is when the sputum contains 1–9 AFB in 100 fields, grade 1+ for 10–99 AFB in 100 fields, grade 2+ if 1–10 AFB per field (check 50 fields), and grade 3+ for more than 10 AFB per field (check 20 fields), respectively [19]. **Body mass index (BMI):** was defined as low when BMI $< 18.5 \, \text{kg/m}^2$, and normal if BMI $18.5–24.99 \, \text{kg/m}^2$ [20]. **Cigarette smoking:** was recorded by asking respondents whether they have ever smoke cigarette in life history. It was dichotomized by 1 (Yes i.e., smoke cigarettes) and 0 (No smoke cigarettes). **Alcohol consumption:** was recorded by asking a respondents whether a respondents have ever drink alcohol or not. It was dichotomized by 1 (Yes i.e., drink alcohol) and 0 (No drink alcohol).

## Data analysis

After data was entered in to epi-data 4.2.0.0, it was exported to Stata 14 for further cleaning, coding, recoding, and analysis. The baseline socio-demographic, behavioural, and clinical features of patients were analysed descriptively. Median time with inter quartile range and mean with Standard Deviation (SD) were used for skewed and normally distributed continues variables, respectively. Texts, tables, and figures were used to present results. The Chi-square assumptions was assessed and valued for categorical explanatory variables.

   Kaplan–Meier survival curve was used to illustrate the median time to culture conversion. Bi-variable and multi-variable Cox proportional hazard model was used to identify the independent predictors of time to smear conversion. The adjusted hazard ratio (AHR) with 95% confidence interval (CI) was used to report the strength of association and statistical significance was declared at p <0.05.

### Ethics approval and consent to participate

The Ethical Review Committee of the University of Gondar approved the study. A permission and support letters were also obtained from the Amhara public health institutions, the management committee of each hospital, and TB ward heads. Since we used secondary data retrospectively, permission letter to access the data was obtained from each study institution as alternative to informed consent. Thus, informed consent was waived as we received the institutional support letter. Information obtained at any course of the study was kept confidential. The data were anonymized and personal identifiers were not included and secured via password in computerized databases to ensure confidentiality.

## Result

### Socio-demographic and Behavioural characteristics of patients

A total of 436 patients were included in the analysis. Majority (84.86%) of the patients were in the age range of 35–54 years. Regarding the residence of the patients, about half of them were residents of urban 225 (51.61%). About one-fifth 91 (20.87%) of the patients had alcohol drinking history at the time of treatment commencement (**Table 1**).

### Clinical characteristics of patients

More than three-fourth 335 (76.83%) of the patients were having a BMI of less than 18.5 kg/m$^2$ at the start of treatment. And above one-fourth 116 (26.61%) of the patients were HIC co-infected during the time they start DR-TB treatment. Moreover, a significant number, 181 (41.51%) of patients had a baseline sputum smear result grading of 3+ (**Table 2**).

### Sputum smear status and conversion time

A total of 436 patients had positive sputum smear microscopy result at the start of the study. The baseline smear status of the patients, graded as scanty, grade 1+, grade 2+, and grade 3

**Table 1. Socio-demographic characteristics and behavioural among DR-TB patients in North West Ethiopia (N = 436).**

| Characteristics | Frequency (%) | Characteristics | Frequency (%) |
|---|---|---|---|
| Age in years | | Baseline smoking history | |
| < 34 | 284 (65.14) | Yes | 66 (15.14) |
| 35–54 | 129 (29.59) | No | 370 (84.86) |
| > = 55 | 23 (5.28) | Baseline alcohol drinking history | |
| Sex | | Yes | 91 (20.87) |
| Male | 255 (58.49) | No | 345 (79.13) |
| Female | 181 (41.51) | Marital status | |
| Educational status | | Married | 203 (46.56) |
| Unable to read and write | 184 (42.20) | Single | 233 (53.44) |
| Primary | 114 (26.15) | Residence | |
| Secondary and above | 138 (31.65) | Urban | 225 (51.61) |
| Occupation | | Rural | 211 (48.39) |
| Farmer | 123 (28.21) | | |
| Housewife | 69 (15.83) | | |
| Daily labourer | 56 (12.84) | | |
| Employed | 46 (10.55) | | |
| Private | 142 (32.57) | | |

**Table 2. Clinical characteristics of patients and treatment related factors among DR-TB patients in North West Ethiopia (N = 436).**

| Characteristics | Frequency (%) | Characteristics | Frequency (%) |
|---|---|---|---|
| Body mass index (BMI) in kg/m$^2$ | | **Adverse drug event** | |
| <**18.5** | 335 (76.83) | Yes | 281 (64.45) |
| **18.5–24.9** | 101 (23.17) | No | 155 (35.55) |
| HIV co-infection | | **Treatment interruption** | |
| **Yes** | 116 (26.61) | Yes | 60 (13.76) |
| **No** | 320 (73.39) | No | 376 (86.24) |
| TB registration group | | **Functional status** | |
| **New** | 58 (13.30) | Working | 151 (34.63) |
| **Relapse** | 63 (14.45) | Ambulatory | 216 (49.54) |
| After failure of treatment | 315 (72.25) | Bed ridden | 69 (15.83) |
| Type of regimen | | *Comorbid conditions | |
| **E-Z-CM-LFX-ETO-CS** | 96 (22.02) | Yes | 69 (15.83) |
| **Z-CM-LFX-ETO-CS** | 52 (11.93) | No | 367 (84.17) |
| **Z-CM-LFX-PTO-CS** | 288 (66.06) | **Baseline sputum smear result grading** | |
| History of treatment for 1$^{st}$ line anti-TB treatment | | 1–9 bacili | 81(18.58) |
| Yes | 380 (87.16) | 1+ | 91(20.87) |
| **No** | 56 (12.84) | 2+ | 83 (19.04) |
| | | 3+ | 181 (41.51) |

*comorbid conditions other than HIV such as diabetes mellitus and Hypertension; E- Ethambutol, Z- Pyrazinamide, CM- Capreomycin, ETO-Ethionamide, PTO-prothionamide, Cs- Cycloserine, LFX- Levofloxacin.

+ was reported as 18.58%, 20.87%, 19.04%, and 41.51%, respectively. From the 436 patients with sputum smear positive at baseline, 351 (80.5%) of them converted sputum smear at a median time of 48 (IQR: 30–78) days. The median time to smear conversion was 61 (95% CI: 36, 73) days among patients with comorbid conditions and 44 (95% CI: 38, 54) days among patients with no comorbid conditions. Similarly, the median time to smear conversion was 59 (95% CI: 42, 74) and 44 (95% CI: 37, 54) days among patients who had and had no history of alcohol drinking, respectively. Moreover, patients who experienced an adverse drug event were smear converted at the median time of 54 (IQR: 41–60) days and those with no adverse drug event were converted at a median time of 40 (IQR: 35–53) days.

The overall incidence rate of sputum smear conversion was 1.93 per 100 Person-Month (PM) (95% CI: 1.69, 2.22) with 11787 total observations. The proportion of sputum smear conversion at 30, 60, 90, and 120 days of treatment initiation was 90%, 42%, 19%, and 9%, respectively (**Fig 2**).

## Predictors of time to smear conversion

The different socio-demographic, behavioural and clinical factors were considered as predictors for smear conversion in the analysis. Accordingly, residence, marital status, history of alcohol drinking, registration group, type of regimen at baseline, history of treatment for 1$^{st}$ line anti-TB treatment, adverse drug event, and baseline sputum smear result grading were found to have a p-value of less than 0.25 in the bi-variable cox regression model and thus considered for the multi-variable analysis. Accordingly, only history of alcohol consumption [AHR = 0.66 (0.50, 0.87)] was found to delay significantly the rate of sputum smear conversion. After controlling for the confounding effect of a number of variables as indicated in the table below, only history of alcohol drinking at baseline was found to be the significant predictor of sputum smear conversion (**Table 3**).

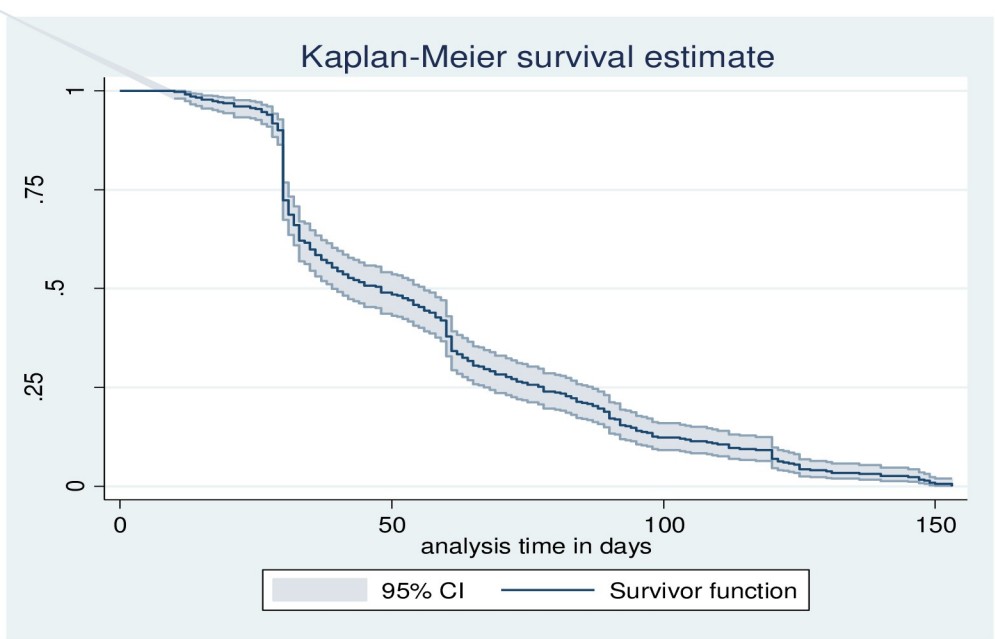

**Fig 2. The Kaplan Meier survivor function for the probability of smear conversion among DR-TB patients in North West Ethiopia (N = 436).**

**Table 3. Bi-variable and multi-variable cox regression analysis of the predictors for sputum smear conversion among DR-TB patients in North West Ethiopia (N = 436).**

| Variables | Categories | Event (%) | Censored (%) | CHR 95% CI | AHR 95% CI |
|---|---|---|---|---|---|
| **Residence** | Rural | 170 (80.57) | 41 (19.43) | 1 | 1 |
| | Urban | 181 (80.44) | 44 (19.56) | 0.87 (0.71,1.08) | 1.19 (0.95, 1.48) |
| **Marital status** | Single | 158 (79.79) | 40 (20.21) | 1 | 1 |
| | Married | 193 (81.09) | 45 (18.91) | 0.85 (0.69,1.06) | 1.23 (0.98, 1.54) |
| **History of alcohol drinking at baseline** | No | 277 (80.29) | 68 (19.71) | 1 | 1 |
| | Yes | 74 (81.32) | 17 (18.68) | 0.84 (0.64,1.09) | **0.66 (0.50, 0.87)** |
| **TB-registration group at baseline** | treatment after failure | 263 (83.49) | 52 (16.51) | 0.64 (0.46,0.89) | 1 |
| | New | 42 (72.41) | 16 (27.59) | 1 | 1.21 (0.55, 2.63) |
| | Relapse | 46 (73.02) | 17 (26.98) | 0.81 (0.53,1.24) | 1.24 (0.89, 1.72) |
| **Type of regimen at Baseline** | Z-CM-LFX-PTO-CS | 225 (78.13) | 63 (21.87) | 1 | 1 |
| | E-Z-CM-LFX-ETO-CS | 84 (87.5) | 12 (12.5) | 0.75 (0.58, 0.96) | 0.81 (0.61, 1.09) |
| | Z-CM-LFX-ETO-CS | 42 (80.77) | 10 (19.23) | 0.66 (0.47, 0.92) | 0.72 (0.51, 1.01) |
| **1st line anti-TB treatment history** | No | 41 (73.21) | 15 (26.79) | 1 | 1 |
| | Yes | 310 (81.58) | 70 (18.42) | 0.67 (0.48, 0.93) | 0.96 (0.69, 1.35) |
| **Adverse drug event** | No | 116 (74.84) | 39 (25.16) | 1 | 1 |
| | Yes | 235 (83.63) | 46 (16.37) | 0.80 (0.64, 1.02) | 1.19 (0.94, 1.50) |
| **Baseline sputum smear result grading** | 1–9 bacili | 65 (80.25) | 16 (19.75) | 1 | 1 |
| | 1+ | 81 (89.01) | 10 (10.99) | 0.79 (0.57,1.09) | 0.70 (0.57, 1.12) |
| | 2+ | 66 (79.52) | 17 (20.48) | 0.82 (0.58, 1.16) | 0.81 (0.56, 1.16) |
| | 3+ | 139 (76.80) | 42 (23.20) | 0.82 (0.61, 1.10) | 0.78 (0.57, 1.08) |

## Discussion

In this study, from a total of 436 patients with sputum smear positive at baseline, 351 (80.5%) of them converted sputum smear at a median time of 48 (IQR: 30–78) days. Tuberculosis patients on anti-tubercular patients are expected to convert the positive sputum smear to negative result at 60 days. As such, the patients in our study had delayed sputum smear conversion than the expected time of conversion.

In this study, the median time of smear conversion was shorter compared to some studies conducted elsewhere [7, 9–11]. The participants' socio-demographic, time difference across studies, and other factors, the study setting could have been responsible for the observed differences in the median time of sputum smear conversion across studies. For instance, one of the studies cited above was a longitudinal study conducted from 2010 to 2012 in which it was relatively earlier than our study. Due to the time gap, there could be numerous transitions in the educational status of the population, people might have more access to the health care and more awareness about the importance of adhering to the treatment thereby they will have short time of smear conversion.

On the other hand, the median time of smear conversion in our study was found to be longer compared to previous studies done in different countries [4, 16, 21]. One of the possible reason for such discrepancy could be that some of the hospitals included in this study had no facility for culture testing during the earlier years of service provision so that they collect and send sputum to the regional health bureau and to other hospitals. This, in turn, might result in delayed reporting or sometimes missing of the result might have occurred thereby longer smear conversion date might be reported. In this study, a number of factors known to affect the rate of sputum smear conversion were evaluated among DR-TB patients. Accordingly, only history of alcohol consumption [**AHR = 0.66 (0.50, 0.87)**] was found to delay significantly the rate of sputum smear conversion. Previous evidences have also indicated that alcohol drinking has effect on sputum and culture conversion, treatment response and thereby treatment outcome of TB/DR-TB patients [12, 22–25]. For example, a multi-centre study conducted in five countries including Peru, Estonia, the Philippines, Latvia, and Russia noted that alcohol drinking resulted in delayed smear/culture conversion time [26].

There could be different justifications for the delayed smear conversion in relation to alcohol drinking. For instance evidence indicated that alcohol consumption is associated with low Pharmacodynamics or delayed effectiveness the anti-TB drugs among alcohol users and thereby delayed smear conversion rate and poor treatment response [27].

There have been established facts about the impact of alcohol use on the immune system of DR-TB patients during the course of anti- TB treatment and its outcome. As such, alcohol consumption was known to reduce macrophage response to immune system modifiers such as cytokines, including interleukin-6 (IL-6), IL-1β, TNF-α, and IL-8 and to prevent the protective effect exerted by the cytokines [28–30]. Moreover, history of alcohol exposure may suppress the capacity of monocytes to produce cytokines, which directly inhibit bacterial growth [22, 31, 32]. Finally, alcohol consumption leads to inhibition of the antigen-specific T-cell activation so that the Th2 population (humoral immunity) dominates the Th1 population (cell-mediated immunity which is responsible for overcoming TB infection) [33, 34]. This shift disturbs a balance between the two basic types of immune system, compromising the immune defense that results in delayed smear conversion and protracted course of treatment thereby poor treatment outcome as a result of alcohol exposure [35–37].

Similarly, the other reason for delayed smear conversion could be related to poor nutritional status, and the direct toxic effects of alcohol on the immune system and poorer adherence to anti-tuberculosis treatment [38, 39]. Also, alcoholism causes drug resistance by

decreasing the immunity status and delay sputum smear conversion among DR-TB patients [40].

Though we conducted this study in different sites and across couples of years which will be mentioned as strength, there has been some limitation in our study. Since the study was based on secondary data, it was not possible to have data on the frequency and type of the alcohol that patients have drunk to further investigate the relationship with the smear conversion rate and the treatment outcome. Moreover, data regarding the radiographic characteristics which has important implication of the rate of smear conversion was not recorded. We however, tri- angulate the DR-TB registration book, the patients' medical chart, and the computerized data record to extract the required data.

## Conclusion

In our study, the median time of sputum smear conversion was with in the expected time frame of conversion. History of alcohol consumption was found to delay significantly the rate of sputum smear conversion.

## Supporting information

**S1 Data.**
(XLS)

## Acknowledgments

It was our pleasure to acknowledge the University of Gondar Specialized hospital for giving permission to conduct this research. We do wish to extend our gratitude to supervisors and data collectors.

## Author Contributions

**Data curation:** Mehari Woldemariam Merid, Getahun Molla Kassa.

**Formal analysis:** Mehari Woldemariam Merid.

**Methodology:** Mehari Woldemariam Merid, Atalay Goshu Muluneh, Getahun Molla Kassa.

**Software:** Mehari Woldemariam Merid.

**Validation:** Mehari Woldemariam Merid, Atalay Goshu Muluneh, Getahun Molla Kassa.

**Visualization:** Mehari Woldemariam Merid, Getahun Molla Kassa.

**Writing – original draft:** Mehari Woldemariam Merid, Atalay Goshu Muluneh, Getahun Molla Kassa.

**Writing – review & editing:** Mehari Woldemariam Merid, Atalay Goshu Muluneh, Getahun Molla Kassa.

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
