## [Decision Letter · Decision Letter 0]

13 Dec 2021

PONE-D-21-34232Alcohol drinking delays the rate of sputum smear conversion among DR-TB patients in northwest Ethiopia; A retrospective follow-up studyPLOS ONE

Dear Dr. Merid,

Thank you for submitting your manuscript to PLOS ONE. After careful consideration, we feel that it has merit but does not fully meet PLOS ONE’s publication criteria as it currently stands. Therefore, we invite you to submit a revised version of the manuscript that addresses the points raised during the review process.

We look forward to receiving your revised manuscript.

Kind regards,

Mao-Shui Wang

Academic Editor

PLOS ONE

Journal Requirements:

- https://www.ncbi.nlm.nih.gov/pmc/articles/PMC2796667/?tool=pmcentrez&renderty=

In your revision ensure you cite all your sources (including your own works), and quote or rephrase any duplicated text outside the methods section. Further consideration is dependent on these concerns being addressed."

a. If there are ethical or legal restrictions on sharing a de-identified data set, please explain them in detail (e.g., data contain potentially sensitive information, data are owned by a third-party organization, etc.) and who has imposed them (e.g., an ethics committee). Please also provide contact information for a data access committee, ethics committee, or other institutional body to which data requests may be sent.

b. If there are no restrictions, please upload the minimal anonymized data set necessary to replicate your study findings as either Supporting Information files or to a stable, public repository and provide us with the relevant URLs, DOIs, or accession numbers. For a list of acceptable repositories, please see http://journals.plos.org/plosone/s/data-availability#loc-recommended-repositories.

Additional Editor Comments:

Most comments focused on How to improve your content. Please follow the advice, and make your responses carefully. In addition, if you disagree with them, please show your difficulties for them, or cite references to support your opinion.

Reviewers' comments:

Reviewer's Responses to Questions

**Comments to the Author**

1. Is the manuscript technically sound, and do the data support the conclusions?

Reviewer #1: No

Reviewer #2: Yes

2. Has the statistical analysis been performed appropriately and rigorously? 

Reviewer #1: No

Reviewer #2: No

3. Have the authors made all data underlying the findings in their manuscript fully available?

Reviewer #1: No

Reviewer #2: Yes

4. Is the manuscript presented in an intelligible fashion and written in standard English?

Reviewer #1: Yes

Reviewer #2: No

5. Review Comments to the Author

Reviewer #1: Comments for Author

Title

• Title is not clear. Title of the study should reflect the outcome variable, population and study area. But the title of this topic lacks the description of outcome variable, population and study area. It needs modifications?

• I believe the main reason to put title in this way is due to the finding of your study. Only one factor is significantly associated with outcome variable. If so, what was new with your findings. Even the median time is within the expected time, what is the significance of this study for the existing knowledge/literature?

• What mean by retrospective follow up study?

Abstract

• The background section of abstract lacks a gap ( a gap in literature, gap in outcome measurement, a gap population, a gap in methodology….)

• Include the aim of the study/objective at the last paragraph Background

Methods

• Add “s” on method

• I think the method section lacks very important components of writing method in abstract, please re-write it (add study area, participants, study area, sample size and sampling techniques)

Results: add “s” on result

Conclusion

• … expected time frame…what was the expected time frame?

• Add recommendation based on the findings of your study. If your finding lack recommendation, your finding lacks signific

Introduction

• Introduction part is very shallow, it is not detail, please add by considering different literatures that focus on this area. What is the magnitude of the problem, where is the gap, what is the significance of conducting this study? If your introduction is not well stated, there is no way to convince the reader for the presence of the problem.

• The second paragraph of introduction cited with one reference, please revise the paragraph

• The third and fourth paragraph of introduction lacks references????

• Overall the introduction part need major revisions

Methods

• What is retrospective follow study?

• Line 89 and 90 of method part is not clear, please re-phrase it.

• Why u used September 2010 as baseline for study period?

• Second paragraph requires a references

• Operational definition need references

• Please explain method of model building

Result

• Briefly write how 436 patients cards included in the final analysis

• Adverse drug effect- what are the lists of ADE

• Scanty, grade 1+, grade 2+, and grade 3+ need operational definition

• What was the proportion censored outcome (for each censored categories)

• What was the incidence rate of smear conversion and person time observation?

• The final table need revision

Discussion

In second paragraph of discussion you stated that socio-demographic and other factors are the reason for discrepancy. What are the socio-demographic and other factors that contribute for the differences and how they could contributes for observed discrepancy?

How sample size increase or decrease the median time to sputum conversion. This is an awkward statement. Or clearly put justification to convince the reader.

Overall, the discussion section is very shallow and it needs major modification. It seems a report of some preliminary data, not the study conducted with advanced statistics.

Conclusion and recommendation

• In our study, the median time of sputum smear conversion was with in the expected time frame of conversion…what means by expected time frame of conversion. If it is with expected time frame what is new with this study?

• Please add the recommendation based your findings

Reviewer #2: Many thanks for the opportunity to review this paper. There is a gap for this issue, especially in Africa on the one hand, with little publications with a whole lot of purpose. The manuscript described a technically sound piece of scientific research with data that supports the conclusions. It has been conducted rigorously, with appropriate controls, replication, and sample sizes. The conclusions are also drawn appropriately based on the data presented. They presented in an intelligible fashion and written in Standard English. However, the following minor issues needs the authors attention

Strength of this

Study could be like:

1.What special approach did you use to increase the quality this

paper?

2.What special characters or features or components did you

include in this study?

3. Since you used secondary data, what special measures did you

take to reduce the limitation associated with secondary data?

If you have relevant answer to these or similar questions you can

mention as strength of this study

Methods and Materials: Sounded, great

1. why you selected the study period between from September 2010 and December, 2017

2. Are you sure all data (i.e Baseline socio-demographic, behavioral, and clinical information can be accessed from medical record reviews? Since it is retrospective cohort study

Result: well narrated

Discussion: Well expressed, however, it will be good if you include the clinical implication of the finding

In conculsion: “In our study, the median time of sputum smear conversion was within the expected time frame of conversion” what is the parameter to concluded median time of sputum smear conversion was within the expected time frame?

6. PLOS authors have the option to publish the peer review history of their article (what does this mean?). If published, this will include your full peer review and any attached files.

Reviewer #1: No

Reviewer #2: No

---

## [Author Response · Author response to Decision Letter 0]

24 Jan 2022

Date: January 17, 2022

Rebuttal letter

PONE-D-21-34232

Alcohol drinking delays the rate of sputum smear conversion among DR-TB patients in northwest Ethiopia; A retrospective follow-up study 

Mehari Woldemariam Merid

To PLOS ONE

Dear all,

We the authors of this manuscript are pleased to thank the journal editors and the reviewers for revising the manuscript and giving your valuable and constructive comments and suggestions that help to improve the manuscript. We have made a rigorous revision of the manuscript as per your questions and comments. We have included the point by point response in the table below framed as editors’ comment/question and authors’ response. We have made a severe revision on the entire manuscript that we believe had merit in improving the manuscript and attached it as tracked change and clean version separately. We are happy to receive additional revision if any that would have merit in improving the manuscript.

Editor comments Authors Response 

Editors comment/suggestion 

1. We noticed you have some minor occurrence of overlapping text with the following previous publication(s), which needs to be addressed:

https://www.ncbi.nlm.nih.gov/pmc/articles/PMC2796667/?tool=pmcentrez&renderty=

In your revision ensure you cite all your sources (including your own works), and quote or rephrase any duplicated text outside the methods section. Further consideration is dependent on these concerns being addressed Thank you for your feed back

As per your comment, we made detail revision on the overlapping text and we paraphrased the text. We also include citations for all the sources we used. This is noted in the tracked change file. Page 3

If you are reporting a retrospective study of medical records or archived samples, please ensure that you have discussed whether all data were fully anonymized before you accessed them and/or whether the IRB or ethics committee waived the requirement for informed consent. If patients provided informed written consent to have data from their medical records used in research, please include this information. Thank you very much for your comment!

This study was solely based on the retrieval of data from the medical charts of DR-TB patients retrospectively (based on secondary data only). Since we used secondary data retrospectively, permission letter to access the data was obtained from each study institution as alternative to informed consent. Thus, informed consent was waived as we received the institutional support letter.

The patient’s data confidentiality was maintained by omitting direct and indirect potential identifiers from the data collection tool. Lines 162-170

When you resubmit, please ensure that you provide the correct grant numbers for the awards you received for your study in the ‘Funding Information’ section. Thank you again

As clearly mentioned in the revised manuscript, this study has no any funding body.

a. If there are ethical or legal restrictions on sharing a de-identified data set, please explain them in detail (e.g., data contain potentially sensitive information, data are owned by a third-party organization, etc.) and who has imposed them (e.g., an ethics committee). Please also provide contact information for a data access committee, ethics committee, or other institutional body to which data requests may be sent.

b. If there are no restrictions, please upload the minimal anonymized data set necessary to replicate your study findings as either Supporting Information files or to a stable, public repository and provide us with the relevant URLs, DOIs, or accession numbers. For a list of acceptable repositories, please see http://journals.plos.org/plosone/s/data-availability#loc-recommended-repositories.

We will update your Data Availability statement on your behalf to reflect the information you provide. Thank you very much for your feedback! And we apologies for the inconvenience.

Since there is no legal restrictions on sharing the data supporting the findings of the current study, we have uploaded the minimal anonymized data set as supporting information files.

5. Your ethics statement should only appear in the Methods section of your manuscript. If your ethics statement is written in any section besides the Methods, please move it to the Methods section and delete it from any other section. Please ensure that your ethics statement is included in your manuscript, as the ethics statement entered into the online submission form will not be published alongside your manuscript. Thank you again for your critique.

Initially, we were placed the ethics statement under the declaration section of the manuscript as per the submission guideline of PLOS One. Now, we moved it in the method section as per your recommendation. Lines 162-170

Reviewer 1 

Questions/Comments Response

Title

1. Title is not clear. Title of the study should reflect the outcome variable, population and study area. But the title of this topic lacks the description of outcome variable, population and study area. It needs modifications? We really appreciate the reviewer’s insight. 

We now made the appropriate modifications on the revised version of the manuscript

As it currently stands, “Alcohol drinking delays the rate of sputum smear conversion among DR-TB patients in northwest Ethiopia; A retrospective follow-up study”

We thought it possesses the information you raised.

Outcome variable- rate of sputum smear conversion

Population- among DR-TB patients

Study area- in northwest Ethiopia

The details are just included in the method section.

We can modify it as;

Time to sputum smear conversion and its predictors among DR-TB patients in northwest Ethiopia; A retrospective follow-up study”

2. I believe the main reason to put title in this way is due to the finding of your study. Only one factor is significantly associated with outcome variable. If so, what was new with your findings. Even the median time is within the expected time, what is the significance of this study for the existing knowledge/literature? Thank you very much again!

Perfect, we prefer to write the title based on the finding from the study, to make emphasis on factors affecting the rate of sputum smear conversion. Accordingly, alcohol drinking was the only factor found to statistically determine the outcome variable. 

As `a researcher, one may not anticipate the finding in favor of his/her wishes. We did the research and it was fortunate that the median time of smear conversion was with in the expected time of conversion as per the protocol (with in two month of treatment commencement). And among the factors considered, only one (alcohol drinking) was remained significant predictors after controlling the confounder variables.

3. What mean by retrospective follow up study? Thank you again!

A retrospective follow up study meant to refer that an event (smear conversion in this study) has been occurred and recorded after being followed for certain period of time historically. The patients have been followed for different time period till smear conversion staring from the time of treatment commencement.

Abstract

1. The background section of abstract lacks a gap ( a gap in literature, gap in outcome measurement, a gap population, a gap in methodology….)

 Thank you for the issue you raised!

Although evidences regarding the factors affecting the sputum smear conversion are available on drug susceptible TB patients, there is dearth of literature about smear conversion and its predictors among DR-TB patients in the study setting. Lines 28-30

2. Include the aim of the study/objective at the last paragraph Background Thank you for the comment

We have included the aim of the study as per your recommendation;

This study has therefore aimed at estimating the median time of sputum smear conversion and to determine its predictors. Lines 32-33

Methods

1. Add “s” on method

 Thank you again 

 “s” is added

2. I think the method section lacks very important components of writing method in abstract, please re-write it (add study area, participants, study area, sample size and sampling techniques) Thank you again for the comment.

As per your comment, we have modified it as follows;

This was a retrospective follow-up study conducted among DR-TB patients registered for second-line anti-TB treatment in the four hospitals of Amhara regional state, Northwest Ethiopia. Of all patients enrolled to DR-TB treatment in the study setting from 2010 to 2017, 436 patients have been include for this study who fulfilled the eligibility criteria.

Results: 

1. add “s” on result Thank you!

Corrected 

Conclusion

1. … expected time frame…what was the expected time frame? Thank you again!

The expected time of conversion was to mean receiving smear negative result within two months of treatment commencement as per the protocol

2. Add recommendation based on the findings of your study. If your finding lack recommendation, your finding lacks significance Thank you for the comment

DR-TB patients are strongly advised to avoid alcohol consumption.

Introduction

1. Introduction part is very shallow, it is not detail, please add by considering different literatures that focus on this area. What is the magnitude of the problem, where is the gap, what is the significance of conducting this study? If your introduction is not well stated, there is no way to convince the reader for the presence of the problem.

 Thank you dear reviewer. 

The introduction part is currently updated as per your recommendation in the revised manuscript. Page 3-4.

The significance of the study is also written as;

Patients with non-conversion or prolonged time to sputum smear conversion may require close monitoring, longer hospitalization and protracted intensive treatment which results in unfavorable treatment outcomes (12-15) . From the public health point of view, reducing the time to sputum smear conversion is an important infection control measure (16). Hence, shortening the time to sputum conversion is desirable to reduce the likelihood of mycobacterial transmission. Moreover, sputum smear conversion is often used by clinicians to determine the duration of treatment for drug resistant tuberculosis patients. 

2. The second paragraph of introduction cited with one reference, please revise the paragraph Thank you again

The reverence is changed.

3. The third and fourth paragraph of introduction lacks references???? Thank you again dear reviewer

Now, we have included citation in the revised manuscript.

 4. Overall the introduction part need major revisions Thank you 

We made a severe revision of the introduction section as per your recommendation. 

Methods

1. What is retrospective follow study?

 Thank you dear reviewer

By retrospective follow-up study, we used it to refer the historical nature of the study where we included patients who have been enrolled to treatment couples of years ago (2010 to 2017).

2. Line 89 and 90 of method part is not clear, please re-phrase it. Thank you dear reviewer

There was an error of text duplication. It is now removed/corrected.

3.Why u used September 2010 as baseline for study period? Thank you dear reviewer

September 2010 was the starting time for DR-TB treatment in most hospitals of the region (study setting). We therefore prefer to include all patients since the treatment commencement date.

4. Operational definition need references Thank you dear reviewer

We added references for the reaming terms operationally defined in the revised manuscript.

5. Please explain method of model building Thank you dear reviewer 

We have just tested the assumption for proportional hazard and was found to satisfy for the data and we fitted the Cox proportional hazard model. We then fitted the model with the variables we have after which a variable having 0.2 p-values in the bi-variable analysis was considered for the multi-variable analysis.

Result

1. Briefly write how 436 patients cards included in the final analysis

 Thank you dear reviewer

This has been clearly explained by the diagram presented in figure 1.

2. Adverse drug effect- what are the lists of ADE Thank you dear reviewer for your valuable comment

In our study, we have included major adverse drug events among DR-TB patients including the following; Acute kidney injury, electrolyte disturbance such as Hypokalemia, Hepatotoxicity, acute psychosis and major gastro-intestinal problems that require treatment.

3. Scanty, grade 1+, grade 2+, and grade 3+ need operational definition Thank you dear reviewer

We have now operationally defined sputum smear grading as follows: 

Sputum smear was graded as scanty, 1+, 2+, and 3+. Scanty is when the sputum contains 1-9 AFB in 100 fields, grade 1+ for 10-99 AFB in 100 fields, grade 2+ if 1-10 AFB per field (check 50 fields), and grade 3+ for more than 10 AFB per field (check 20 fields), respectively (19). Lines 141-144

4. What was the proportion censored outcome (for each censored categories) Thank you dear reviewer for the comment

As per your comment we now included the proportion in the revised manuscript.

5. What was the incidence rate of smear conversion and person time observation? Thank you dear reviewer

The objective of this study was to estimate the median time of smear conversion and determine the predictors. We, however, added the incidence rate of smear conversion in the revised manuscript as follows;

The overall incidence rate of sputum smear conversion was 1.93 per 100 Person-Month (PM) (95% CI: 1.69, 2.22) with 11787 total observations.

6. The final table need revision Thank you dear reviewer

As per, your recommendation, the table is update in the revised manuscript. Pages 9-11 table 3

Discussion

1. In second paragraph of discussion you stated that socio-demographic and other factors are the reason for discrepancy. What are the socio-demographic and other factors that contribute for the differences and how they could contributes for observed discrepancy?

How sample size increase or decrease the median time to sputum conversion. This is an awkward statement. Or clearly put justification to convince the reader.

Overall, the discussion section is very shallow and it needs major modification. It seems a report of some preliminary data, not the study conducted with advanced statistics.

Conclusion and recommendation

• In our study, the median time of sputum smear conversion was with in the expected time frame of conversion…what means by expected time frame of conversion. If it is with expected time frame what is new with this study?

• Please add the recommendation based your findings Thank you dear reviewer

As per your valuable comment, we have made a detail revision on the discussion section and improved it in the revised manuscript.

Some of the modifications are as follows.

………For instance, one of the studies cited above was a longitudinal study conducted from 2010 to 2012 in which it was relatively earlier than our study. Due to the time gap, there could be numerous transitions in the educational status of the population, people might have more access to the health care and more awareness about the importance of adhering to the treatment thereby they will have short time of smear conversion. 

…….One of the possible reason for such discrepancy could be that some of the hospitals included in this study had no facility for culture testing during the earlier years of service provision so that they collect and send sputum to the regional health bureau and to other hospitals. This, in turn, might result in delayed reporting or sometimes missing of the result might have occurred thereby longer smear conversion date might be reported.

Page….

As `a researcher, one may not anticipate the finding in favor of his/her wishes. We did the research and it was fortunate that the median time of smear conversion was with in the expected time of conversion as per the protocol (with in two month of treatment commencement). 

Reviewer 2 

1. Strength of this

Study could be like:

1.What special approach did you use to increase the quality this

paper?

2.What special characters or features or components did you

include in this study?

3. Since you used secondary data, what special measures did you

take to reduce the limitation associated with secondary data?

If you have relevant answer to these or similar questions you can

mention as strength of this study Thank you very much for commending our work!

We have made our maximum effort to extract data by integrating the patients’ unique DR-TB number with the computerized data base so as not to miss the data not registered in the medical charts. We just triangulate the DR-TB registration book, the patients’ medical chart, and the computerized data record to extract the required data.

2. Methods and Materials: Sounded, great

1. why you selected the study period between from September 2010 and December, 2017

2. Are you sure all data (i.e Baseline socio-demographic, behavioral, and clinical information can be accessed from medical record reviews? Since it is retrospective cohort study Thank you very much for your valuable comment.

Most of the DR-TB treatment initiating centers started enrolling patients since 2010 in the region. Hence, to ensure representativeness and obtain adequate sample, we prefer to study from 2010 to 2017. Some variables were not recorded in the medical charts. For those variables, we tried to search from the DR-TB registration and computerized data base.

3. Result: well narrated

Discussion: Well expressed, however, it will be good if you include the clinical implication of the finding

In conculsion: “In our study, the median time of sputum smear conversion was within the expected time frame of conversion” what is the parameter to concluded median time of sputum smear conversion was within the expected time frame? Thank you again for your valuable comment.

The expected time of conversion was to mean receiving smear negative result within two months of treatment commencement as per the protocol

---

## [Decision Letter · Decision Letter 1]

18 Feb 2022

Alcohol drinking delays the rate of sputum smear conversion among DR-TB patients in northwest Ethiopia; A retrospective follow-up study

PONE-D-21-34232R1

Dear Dr. Merid,

We’re pleased to inform you that your manuscript has been judged scientifically suitable for publication and will be formally accepted for publication once it meets all outstanding technical requirements.

Kind regards,

Mao-Shui Wang

Academic Editor

PLOS ONE

Additional Editor Comments (optional):

Reviewers' comments:

Reviewer's Responses to Questions

**Comments to the Author**

1. If the authors have adequately addressed your comments raised in a previous round of review and you feel that this manuscript is now acceptable for publication, you may indicate that here to bypass the “Comments to the Author” section, enter your conflict of interest statement in the “Confidential to Editor” section, and submit your "Accept" recommendation.

Reviewer #1: All comments have been addressed

Reviewer #2: All comments have been addressed

2. Is the manuscript technically sound, and do the data support the conclusions?

Reviewer #1: Yes

Reviewer #2: Yes

3. Has the statistical analysis been performed appropriately and rigorously? 

Reviewer #1: Yes

Reviewer #2: I Don't Know

4. Have the authors made all data underlying the findings in their manuscript fully available?

Reviewer #1: Yes

Reviewer #2: Yes

5. Is the manuscript presented in an intelligible fashion and written in standard English?

Reviewer #1: Yes

Reviewer #2: Yes

6. Review Comments to the Author

Reviewer #1: Dear author, thank you for revising the manuscript. All of my comments have been addressed, and I recommend the journal to consider to this paper for publication.

Reviewer #2: Thank you for give the opportunity to review the paper for entieted "Alcohol drinking delays the rate of sputum smear conversion among DR-TB patients in northwest Ethiopia; A retrospective follow-up study" (PONE-D-21-34232R1).

I appreciate the authors your precious time to providing valuable modification and considering the comments.

All my concern was adresses in this round , and made the manuscript acceptable for publication. Since the authors provided valuable and insightful response that led to possible improvements in the current version that he authors have carefully considered the comments and tried thier best to address every one of them.

And the I would like to to say congratulation one more.

Thanks!

7. PLOS authors have the option to publish the peer review history of their article (what does this mean?). If published, this will include your full peer review and any attached files.

Reviewer #1: No

Reviewer #2: No

---

## [Editor Report · Acceptance letter]

28 Feb 2022

PONE-D-21-34232R1 

Alcohol drinking delays the rate of sputum smear conversion among DR-TB patients in northwest Ethiopia; A retrospective follow-up study 

Dear Dr. Merid:

I'm pleased to inform you that your manuscript has been deemed suitable for publication in PLOS ONE. Congratulations! Your manuscript is now with our production department. 

Kind regards, 

on behalf of

Dr. Mao-Shui Wang 

Academic Editor

PLOS ONE